# Preparation and Characterization of *Bletilla striata* Polysaccharide/Polylactic Acid Composite

**DOI:** 10.3390/molecules24112104

**Published:** 2019-06-03

**Authors:** Renyu Yang, Dongyue Wang, Hongli Li, Yi He, Xiangyu Zheng, Mingwei Yuan, Minglong Yuan

**Affiliations:** National and Local Joint Engineering Research Center for Green Preparation Technology of Biobased Materials, Yunnan Minzu University, Kunming 650500, China; yangrenyu1995@163.com (R.Y.); wangdongyue1992@163.com (D.W.); honglili@vip.163.com (H.L.); heyi_sichuan@163.com (Y.H.); zhengxiangyu1993@163.com (X.Z.); yuanmingwei@163.com (M.Y.)

**Keywords:** *Bletilla striata* polysaccharide, polylactic acid, blend modification, solvent method

## Abstract

Polylactic acid (PLA) is limited in its application due to its high price, high brittleness and low glass-transition temperature. Modification methods are currently used to overcome these shortcomings. In this study, *Bletilla striata* polysaccharide (BSP) was blended with PLA by a solvent method. DMA data showed that the BSP/PLA film had a higher glass-transition temperature, and the glass-transition temperature of the film showed an extreme value of 68 °C when the proportion of the chalk polysaccharide was 0.8%. TG data indicates that the composite film material has good thermal stability. Tensile tests show that the composite film is improved in rigidity and elasticity compared to the pure PLA film. The blending modification of PLA with white peony polysaccharide not only reduces the cost of PLA, but also improves the thermal and mechanical properties of PLA.

## 1. Introduction

Extracted from Radix Paeoniae Alba, *Bletilla striata* polysaccharide (BSP), also known as *Bletilla hyacinthina gumis*, is a polymer with low toxicity and high safety. The main components of BSP are glucose and mannose [1]. The naturally extracted BSP is a powder with a slight sweetness. The powder can dissolve in water to form a viscous solution, which can be used as a film-forming material. As a natural polymer material, BSP exhibits good biocompatibility and biodegradability [2], having excellent application prospects [3,4]. BSP has high safety as a food additive or ingredient, and its unique properties make it widely used in the food industry [5]. Related research shows that using the excellent film-forming properties of BSP, the preparation of fruit coating film preservative can reduce the evaporation of water and achieve the purpose of preservation [6]. BSP has good anti-inflammatory and acid-resistance ability, and is less affected by factors such as pH, inorganic ions and temperature, and can improve the stability of the emulsified product. With the development and clinical needs of modern pharmaceutical technology, BSP is also widely used in pharmacology and clinical practice as mediation therapy, coupling agent, etc. Cai et al. [7] developed a white ultrasonic medical couplant, and compared it with paraffin oil emulsion and Japanese ultrasonic coupling agent, found that the main quality index of white enamel ultrasonic coupling agent exceeds paraffin oil emulsion, which is superior to Japanese products.

Polylactic acid (PLA) is a natural polymer material with a wide range of sources. PLA can be regenerated, is completely biodegradable and nontoxic and it exhibits good biocompatibility and biodegradability [8,9,10]. In addition, PLA has a wide range of applications and is easy to process. It is a green polymer with excellent performance and can blend with other natural polymer materials to form fully biodegradable composites [11,12,13]. With the theme of environmental protection and sustainable development, PLA has attracted increasing attention and has been extensively studied in the fields of industry and agriculture [14,15], biomedicine [16,17] and food packaging [18,19,20]. At present, PLA is considered one of the materials with the most promising development prospects [21]. Ma et al. [22] obtained the PLA/Fe_3_O_4_-AZM microspheres by emulsification-solvent evaporation technology, and the sustained release effect was obvious. Tan et al. [23] obtained a PLA/PCL-PVA-CS-Ag nanofiber auxiliary material by electrospinning technology and achieved good antibacterial effects. Swaroop et al. [24] used polylactic acid as a raw material to prepare magnesium oxide nanoparticle-enhanced biofilms by a solvent casting method. The prepared film is transparent, can shield ultraviolet radiation, and has excellent antibacterial properties. It is an excellent food packaging material. However, the high price of PLA has constrained its development [25]. In addition, other factors such as low heat-resistance, poor hydrophilicity, high brittleness, insufficient elasticity, low degradation rate, varied degradation cycle, low mechanical strength, poor moisture vapor barrier property, low crystallization rate and low thermal stability [26,27,28,29] greatly limit the application scope of PLA. Therefore, physical modification, chemical modification and composite preparation have been carried out to modify PLA materials to improve their performance and reduce cost.

Wu et al. [30] used elastomers and PLA to prepare composites. The obtained materials have the advantages of a short plasticizing time, high melt fluidity, and high tensile elongation at break, but they have the disadvantage of significantly decreased tensile strength. Pan et al. [31] prepared a polyacrylic acid grafted starch/PLA composite using a graft copolymerization and blending technique. The obtained products have significantly increased tensile strength and hydrophilicity but exhibit decreased degradation and thermal performance. Similarly, Wu et al. [32] observed increased tensile strength and elongation at break for a polylactic acid grafted starch/PLA composite but it had a significantly decreased initial thermal decomposition temperature and deteriorated thermal stability. Tao et al. [33] prepared aliphatic polycarbonate (PPC)/PLA composites by solution casting. The obtained materials exhibit an increased biodegradation rate that improves the biodegradability of PLA but also exhibit a lower tensile strength and modulus compared with PLA. As such, the thermal properties of the materials are not improved. Zhang et al. [34] modified PLA with biodegradable hyperbranched polyester amide ester (HBP) by melt blending. The obtained PLA composites have increased tensile strength and elongation at break, excellent toughness but poor thermal stability. Li et al. [35] used tetrabutyl titanate to modify a PLA/starch composite. The obtained materials exhibit enhanced flexibility and degradation performance but show no improvement in thermal properties. Zhang et al. [36] prepared and characterized PLA/cellulose nanocrystalline composites. The obtained materials have increased tensile strength but decreased elongation at break and deteriorated toughness. Feng et al. [37] blended polyurethane with PLA. The obtained materials have the advantages of high elongation at break, high impact strength and high tensile strength but exhibit no improvement in thermal properties. Thus, research has shown that using natural polymer materials such as starch or cellulose to modify PLA can only improve either mechanical properties or thermal properties but not both. Therefore, finding a suitable material that can improve both the mechanical properties and thermal properties of PLA is important. Compared with other natural polymer materials, BSP has the advantages of wide availability, low cost, excellent toughness, and natural degradability. The blending of BSP and PLA can not only reduce the cost but also improve the performance of PLA materials. In this study, BSP is selected as the material for modifying PLA. The results show that the obtained composite exhibits improved thermal properties and improved mechanical properties, which is a new highlight in PLA blend modification research.

In this study, BSP was used as the modified material, PLA was used as the raw material, and 1,4-dioxane and ultrapure water were used as solvents to physically blend PLA and BSP to prepare BSP/PLA composite films. The optimum ratio was deduced to achieve the best modification effect. The glass-transition temperature (T_g_) of the composite film was determined by dynamic thermomechanical analysis (DMA). The crystal morphology was determined by X-ray diffraction (XRD). The thermal decomposition temperature was determined by thermogravimetry (TG). The thermal stability was determined by differential scanning calorimetry (DSC) and TG. The morphologies of the cross section and surface were observed by scanning electron microscopy (SEM). In addition, the mechanical properties of the composite film were tested on a tensile testing machine.

## 2. Results and Discussion

### 2.1. Loss Factor Analysis of DMA

The T_g_ is an excellent indicator of the application range of a material [38]. The higher the Tg is, the better the processing of the material, the wider the application range, and the higher the upper limit of the plasticity [39]. The DSC and DMA data can both reflect the T_g_ of the BSP/PLA composite films. However, because the thermal effect of DMA is more obvious and the DMA data are more convincing, we used DMA to measure the T_g_ of the composite films. To determine the T_g_, two tangent lines were drawn on the tanδ-temperature curve and the abscissa of the intersection of the tangent lines was considered the T_g_.

PLA1, PLA2, PLA3, PLA4, PLA5, PLA6, PLA7, PLA8 and BSP represent the BSP content of 0%, 0.2%, 0.4%, 0.6%, 0.8%, 1%, 1.2%, 1.4% and 100% of the total mass, respectively.

A dynamic thermomechanical analyzer was used to measure the T_g_ of the BSP/PLA composite films with different BSP ratios. Figure 1 shows the variation curve of the wear factor with temperature for various BSP/PLA composite films (plotted with Origin v7.0; OriginLab Corp., Northampton, MA, USA). Figure 1 and Table 1 show the T_g_ of the composite films with various BSP ratios. With increasing BSP ratio, the T_g_ of the composite films first increases, reaches the peak value at 0.8%, and then gradually decreases as the BSP ratio is increased further. This advantageous property was not observed by Wu [30], Pan [31] or Wu [32]. This is because there is a hydrogen bond between BSP and PLA, which hinders the migration and diffusion of the PLA chain, so the molecular chain rigidity increases and the glass transition temperature increases. With the further increase of the BSP ratio, excessive BSP increases molecular chain distance. Therefore, the T_g_ decreases as the BSP ratio is increased further [40].

### 2.2. Energy Storage Modulus Analysis of DMA

Changes in temperature will cause changes in molecules, and changes of related viscoelastic properties such as the energy storage modulus and loss modulus. Figure 2 shows the effects of different ratios of BSP on the system. The results in Figure 2 show that the storage modulus of the composite film is positively correlated with ratio of BSP; i.e., the higher the BSP ratio, the greater the storage modulus of the film. The storage modulus of the two composite films with the highest ratios of BSP is far superior to those of other films at the beginning of the measurement. In addition, when the temperature rises to the T_g_, these two films still show a relatively greater storage modulus than the other films, indicating that the introduction of BSP enhances the elasticity of the material. Compared with the films prepared by Zhang et al. [35], the films prepared in this study have certain advantages in terms of storage modulus and elasticity.

### 2.3. DSC Analysis of Composite Films

The DSC curves of pure PLA and PLA modified with different ratios of BSP are shown in Figure 3, and the obtained parameters are shown in Table 2 (T_c_ is the cold crystallization temperature of the material, ΔH_c_ is the enthalpy of crystallization, T_m_ is the melting temperature, ΔH_f_ is the enthalpy of fusion, and X_c_ is the crystallinity). At 85.5 °C, pure PLA exhibits a very small cold crystallization peak. After the addition of BSP, the position of the melting peak does not substantially change; however, the intensity of the peak is slightly higher than that of pure PLA. The positions of the cold crystallization peaks generally shift toward higher temperatures, indicating that BSP can promote the heterogeneous nucleation of PLA in the temperature range 80–130 °C. However, the shift of the cold crystallization peak toward lower temperatures also indicates that the modified film can only crystallize at higher temperatures, which makes the formation of crystal nuclei difficult. Therefore, the composite films are more difficult to crystallize at lower temperatures than pure PLA. The T_m_ of composite films increases with increasing BSP ratio and are all higher than that of pure PLA, although this increase is not obvious. However, the extent of the increase of the T_m_ shows a trend of first increasing and then decreasing. We speculate that, when the temperature is too high, molecules undergoing intense thermal motion cause the formed nuclei to become unstable; the nuclei are thus susceptible to be affected by the thermal movement of the molecular chain of PLA.

According to the formula, we calculated the crystallinity of each film using the measured enthalpy of fusion and found that the variation trend of crystallinity also first increases and then decreases.

### 2.4. TG Analysis of Composite Films

Figure 4 shows the TG curve and the DTG curve of the composite films under a nitrogen atmosphere. In Table 3, T_initial_ represents the initial decomposition temperature of the material, T_10_ represents the temperature at which the material has undergone 10% weight loss, T_complete_ represents the temperature at which the material completely decomposes, and T_max_ represents the temperature corresponding to highest decomposition rate.

Figure 4 shows that the weight loss of BSP from 25 to 200 °C is mainly due to the loss of free water and bound water. The weight loss between 200 and 350 °C is mainly due to the loss of polysaccharides. The weight loss above 350 °C might be due to the loss of unburned BSP and carbonized PLA. DTG thermograms of a few composite films show increases in the weight-loss rate at approximately 220 and 250 °C, which may be due to the loss of the unevenly mixed BSP. On the basis of the data in Table 3, with the continuous increase of the BSP ratio, both the T_10_ and the T_max_ of the composite films first increase and then decrease; both are higher than those of pure PLA. When the ratio of BSP is 0.8%, the T_10_ and the T_max_ of the composite films both reach the maximum, which are 116.3 °C and 45.9 °C higher than those of the pure PLA, respectively. The increase of T_10_ and T_max_ is due to the formation of hydrogen bonds between BSP and PLA, which hinders the relative movement of PLA molecules when heated. This improves the thermal stability of the PLA. When the ratio of BSP is increased further, the distance between the molecular chains is increased, and the entanglement of the PLA chain is reduced, so the thermal stability is reduced. This is consistent with the change in crystallinity.

Pure PLA has a T_complete_ of 359.3 °C and a T_initial_ of 291.2 °C. With the continuous addition of BSP, the T_complete_ and T_initial_ of the composite films also continue to increase until reaching the peak value (398.5 °C for T_complete_ and 351.2 °C for T_initial_) when the BSP ratio is 0.8%. Afterwards, T_complete_ and T_initial_ both show a decreasing trend; overall, however, they are both higher than the T_complete_ and T_initial_ of PLA: the maximum T _complete_ of the composites is 39.2 °C higher and the maximum T_initial_ is 60 °C higher than those of pure PLA, indicating that the composite films exhibit greater thermal stability than pure PLA. Previous studies on PLA modification mostly improved its mechanical properties, not its thermal properties. The composites obtained in this study have the advantages of improved thermal performance.

### 2.5. XRD Analysis of Composite Films

The XRD patterns of the BSP/PLA composite films are shown in Figure 5. The single crystal diffraction peak of 16.69° corresponds to the α crystal form of polylactic acid, and the single crystal diffraction peak of 18.52° is the diffraction peak of BSP. With increasing BSP ratio, the position of the diffraction peak of the PLA single crystal is significantly shifted to a smaller 2θ angle, and then slightly shift to a larger 2θ angle. According to the Brugg equation 2dSinθ = nλ, it can be found that the crystal plane spacing first increases significantly and then decreases slightly, and the crystal core stacking first loosens and then compacts.

### 2.6. SEM Analysis of Composite Films

The SEM images of the composite films are shown in Figure 6, and the pore sizes are shown in Table 4. The microstructure of the composite films is a loose and porous network structure. Figure 6a–f present SEM images of the films with a BSP ratio 0.4%, 0.6%, 0.8%, 1.0%, 1.2% and 1.4%, respectively. With increasing BSP ratio, the pores of the composite film are firstly increased and then reduced by the influence of the interplanar spacing, which is consistent with the XRD results.

### 2.7. Mechanical Analysis of Composite Films

The tensile strength of the BSP/PLA composite film is shown in Figure 7 and Table 5. The elongation at break of the composite films showed a trend of increasing first and then decreasing, reaching the peak value of 53.14% when the BSP ratio is 0.8%. This peak value is 2.46 times greater than that of pure PLA, indicating improved toughness of the material [41]. This is because when the BSP is small, the compatibility of the material is good [42], and the BSP is uniformly dispersed in the PLA, so the elongation at break increases, when the BSP is large, the partial molecular chain of the PLA may be crosslinked to form a gel point, so the elongation at break will decrease. But the tensile strength of the composite films showed a trend of decreasing first and then increasing, this is because when the BSP is small, the intermolecular interaction force is weak, so the tensile strength is reduced, when the BSP content is high, BSP will crosslink with PLA, and the intermolecular interaction force is strong, so the tensile strength is improved. Finally, the addition of BSP also increases the maximum force of the material by as much as 1.98 times that of pure PLA, indicating that BSP can be used to toughen PLA.

### 2.8. Contact Angle of Composite Films

The smaller the contact angle of the material, the better the hydrophilicity of the material. According to Figure 8 and Table 6, it can be seen that with the increase of BSP, the hydrophilicity of the material is getting better and better.

## 3. Materials and Methods

### 3.1. Materials

BSP was purchased from Xi’an Tianrui Biotechnology Co., Ltd. (Xi′an, China). Activated carbon was purchased from Jiangsu Agnes Environmental Technology Co., Ltd. (Jiangsu, China). Ethanol and 1,4-dioxane were purchased from Tianjin Damao Chemical Reagent Co. (Tianjin, China). PLA was prepared in our own laboratory, and the average molecular weight was 11.0 × 10^4^. A glass plate with an inner diameter of 20 cm × 20 cm was prepared in our laboratory.

### 3.2. Purification of BSP

Fifty grams of extracted BSP was dissolved in 1000 mL of distilled water and continuously stirred with a glass rod. After BSP was completely dissolved, the mixture was filtered with gauze 3 times and the filtrate was collected by vacuum filtration. A small amount of activated carbon was added to the collected filtrate, and the mixture was heated with stirring for 1 h. After hot suction filtration, the obtained BSP solution was heated and concentrated to 400 mL. After the solution cooling, ethanol was slowly added with stirring until the glue was completely precipitated. After setting overnight, the precipitate was collected by suction filtration and dried at 80 °C to yield 35 g of BSP.

### 3.3. Preparation of BSP/PLA Composite Films

The BSP/PLA composite film was prepared by a solvent volatilization technique. Two grams of PLA was added to each of two 100 mL conical flasks. After 50 mL 1,4-dioxane was added, the mixture was stirred for 12 h until complete dissolution of PLA. BSP in amounts of 0%, 0.2%, 0.4%, 0.6%, 0.8%, 1%, 1.2%, 1.4% and 100% of the amount of PLA (referred to hereafter as PLA1, PLA2, PLA3, PLA4, PLA5, PLA6, PLA7, PLA8 and BSP, respectively) was accurately weighed into 1 mL centrifuge tubes, and 0.6 mL of ultrapure water was added to each tube. After BSP was completely dissolved, the solution from one of the centrifuge tubes was added into the conical flask drop by drop and the resultant solution was stirred for 4 h. Afterwards, the mixture was poured onto a flat glass plate sitting on a flat platform. The plate was dried in a wind-free dry environment for 36 h until the solvent had completely evaporated. The dried film, with a thickness range of 50–55 μm, was collected for measurement and subsequent performance evaluation.

### 3.4. Dynamic Thermomechanical Analysis (DMA) Analysis

A dynamic mechanical analyzer (DMA 242 E, Netzsch, Selb, Germany) was used to collect DMA data. First, samples were made into 5 mm × 15 mm specimens. The experiments were conducted via the dynamic stretching method; the initial frequency was 1 Hz, the absolute amplitude was 120 μm, the maximum dynamic force was 1.5 N, the additional static force was 0.05 N, the heating rate was 2 °C/min, and the temperature range was 25–180 °C.

### 3.5. Different Scanning Calorimeter(DSC) Analysis

DSC data were collected using a differential scanning calorimeter (DSC 214 Polyma, Netzsch). A sample of 3–5 mg was weighed and sealed in an aluminum crucible. The N2 flow rate was 30 mL/min, the heating rate was 5 °C/min, the temperature range was 25–200 °C. TG-DSC was used to identify and compare the starting temperature where peaks appeared in the thermogram. The crystallinity formula [43] is as follows:$$X_{c}=\frac{H_{f}{(T}_{m})}{H^{o}{}_{f}{(T}^{o}{}_{m})}\;100\%$$
where $X_{c}$ is the crystallinity of the sample, $H_{f}{(T}_{m})$ is the enthalpy of fusion, and $H^{o}{}_{f}{(T}^{o}{}_{m})$ is the enthalpy of fusion for the sample with complete PLA crystallization, which is 93.0 J/g.

### 3.6. Thermogravimetric(TG) Analysis

Thermogravimetric analysis (TGA) data were obtained using a thermogravimetric analyzer (STA449F31, Netzsch). The heating rate was controlled at 10°C/min under a nitrogen atmosphere. The TGA analysis was carried out between 25 and 500 °C.

### 3.7. X-Ray Diffraction Characterization (XRD)

The samples were subjected to X-ray analysis using a wide-angle X-ray diffractometer (Bruker, Karlsruhe, Germany). A C_u_ target (λ = 0.154 nm) operated at a working voltage 45 kV and working current 150 mA was used; the scanning-angle range was 5–60°, and the scanning rate as 5°/min.

### 3.8. Scanning Election Microscopy (SEM )Analysis

SEM (NOVA NANOSEM-450, FEI, Hillsboro, OR, USA) was used to observe the surface and cross-section of the sample. To effectively observe the surface and cross-section, the sample was first quenched in liquid nitrogen and the cross-section was subjected to a gold spray treatment. The accelerating voltage was 5 kV, and the magnification was 5000×. The pore size was analyzed using the Nano-measurer software.

### 3.9. Mechanical Performance Test

The mechanical properties of the samples were tested using a microcomputer-controlled electronic universal testing machine (CMT 4104, MTS Industrial Systems Co., Inc., Shenzhen, China). First, the sample was made into a 45 mm × 10 mm specimen, and a 50 N sensor was selected for the measurement. The tensile performance test of a plastic film was selected as the measurement mode. The test speed was 100 mm/min. Parallel experiments (seven each) were conducted, and average values were taken.

### 3.10. Contact Angle Test

Measurements were made using a contact angle measuring instrument model OCA200 (Dataphysics, Stuttgart, Germany). A 2 cm diameter sample was mounted on a clean, smooth glass slide and the drop was dropped onto the surface of the film by hanging drop. The contact angle of water on the surface of the film was measured by a contact angle meter, and each sample was measured twice.

## 4. Conclusions

In this study, BSP was used to physically modify PLA to improve its performance. The results show that, when the ratio of BSP is 0.8%, the BSP/PLA composite film exhibits the best performance. DMA shows that the addition of BSP can increase the Tg (to a maximum temperature of 68 °C) and improve the rigidity and elasticity of the materials. DSC demonstrates that the modified PLA exhibits increased crystallinity. TGA indicates that the addition of BSP can increase the thermal decomposition temperature to a maximum of 60 °C. XRD shows that the introduction of BSP does not change the crystalline form. SEM shows that BSP can destroy PLA molecular chains, making it difficult to migrate and diffuse. Mechanical analysis shows that, with the continuous addition of BSP, the maximum elongation at break of the film increases by 2.46 times and the maximum force also increases to a peak value that is 1.98 times that of pure PLA. In this study, the BSP/PLA composite film was prepared and characterized for the first time. The PLA modified by BSP could find applications in the green packaging field.

## Figures and Tables

**Figure 1 molecules-24-02104-f001:**
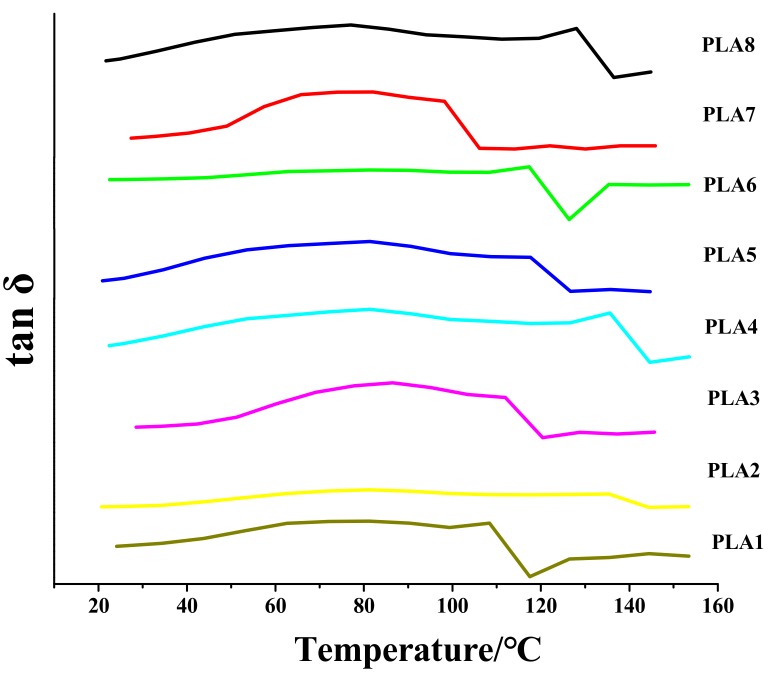
Analysis of DMA Loss Factor of BSP/PLA Composite Films with Different Proportions. Temperature/Loss Factor diagram.

**Figure 2 molecules-24-02104-f002:**
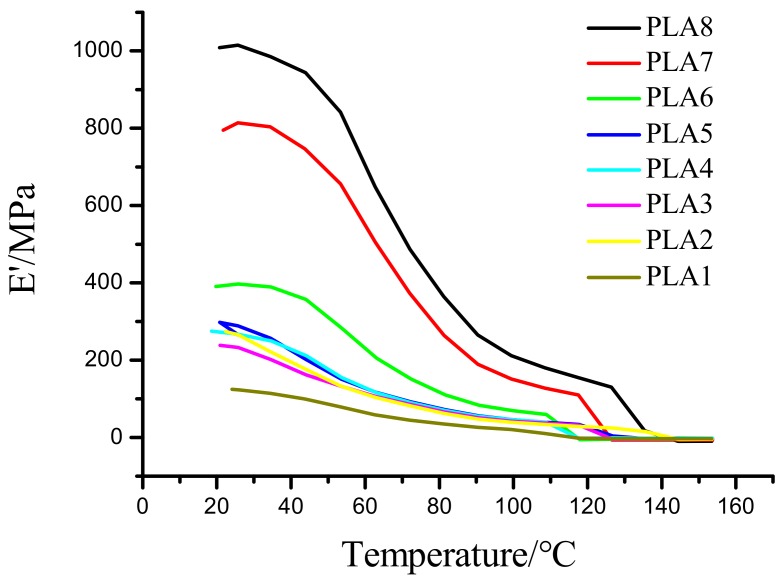
Analysis of Storage Modulus of BSP/PLA Films. Temperature/storage modulus diagram.

**Figure 3 molecules-24-02104-f003:**
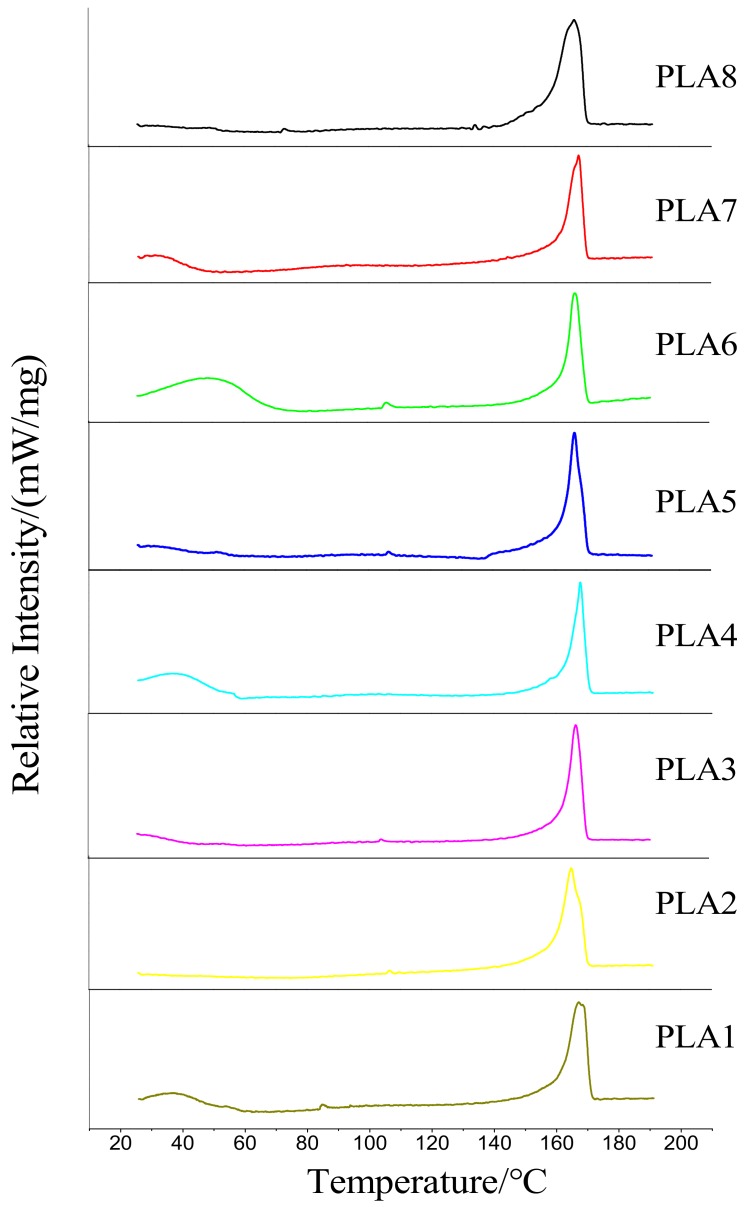
DSC of BSP/PLA Composite Films. Temperature/heat flow diagram.

**Figure 4 molecules-24-02104-f004:**
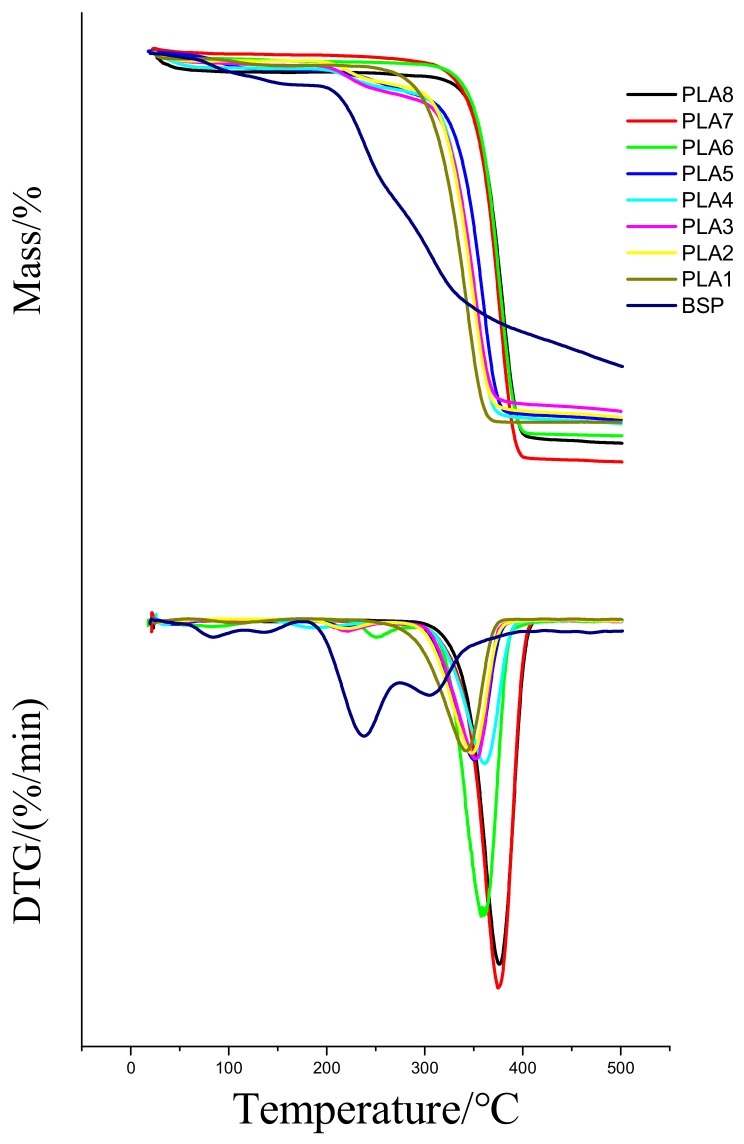
TG and DTG Curves of Composite Films. Temperature/thermogravimetry and Temperature/differential thermogravimetry dargram

**Figure 5 molecules-24-02104-f005:**
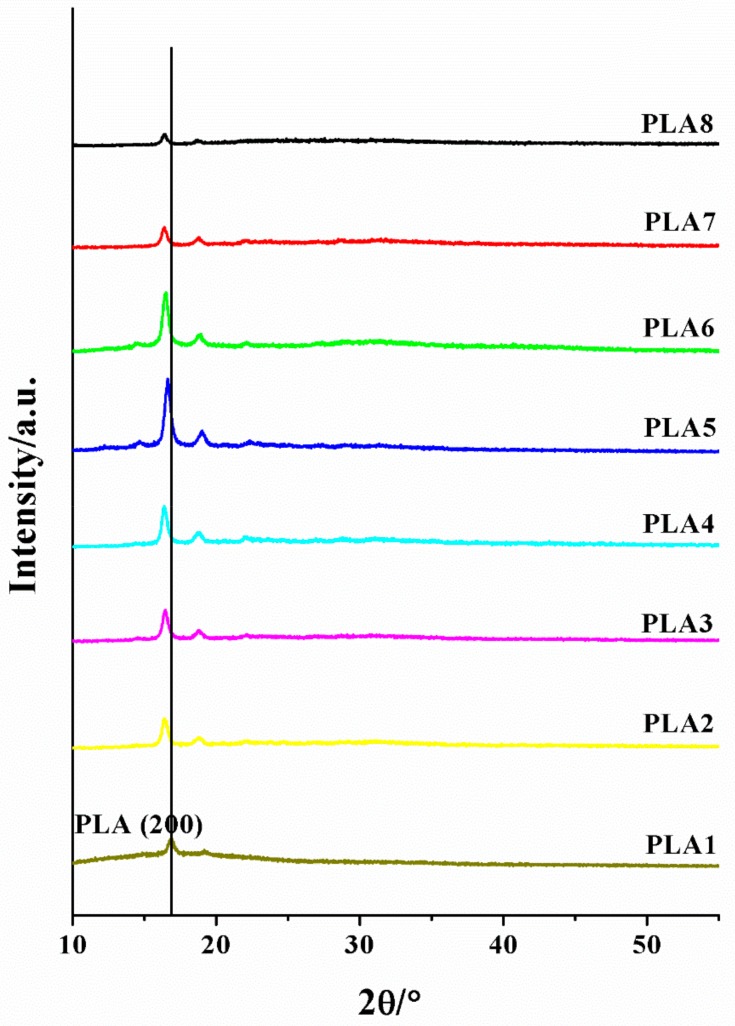
XRD of composite films. 2θ/ Diffraction peak intensity diagram.

**Figure 6 molecules-24-02104-f006:**
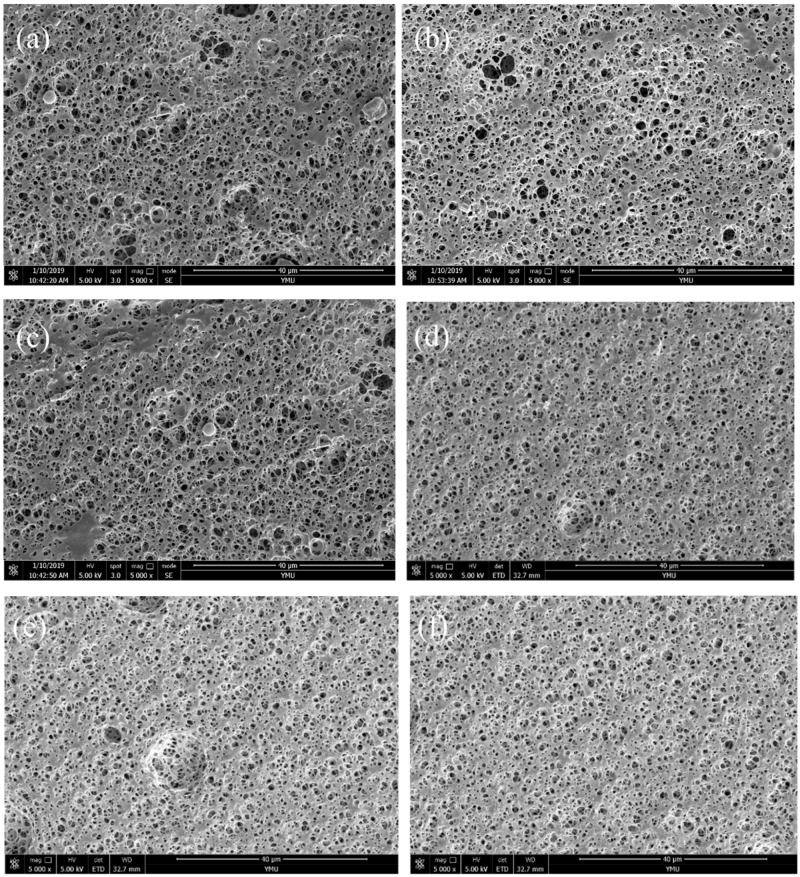
SEM of composite films. (**a**) Represents a BSP ratio of 0.4%; (**b**) represents a BSP ratio of 0.6%; (**c**) represents a BSP ratio of 0.8%; (**d**) represents a BSP ratio of 1%; (**e**) represents a BSP ratio of 1.2%; (**f**) represents a BSP ratio of 1.4%.

**Figure 7 molecules-24-02104-f007:**
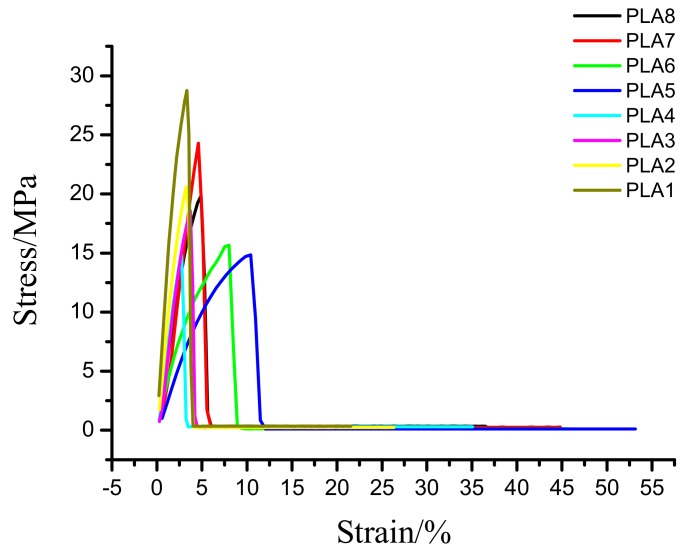
Plot for tensile data. Stress strain diagram

**Figure 8 molecules-24-02104-f008:**
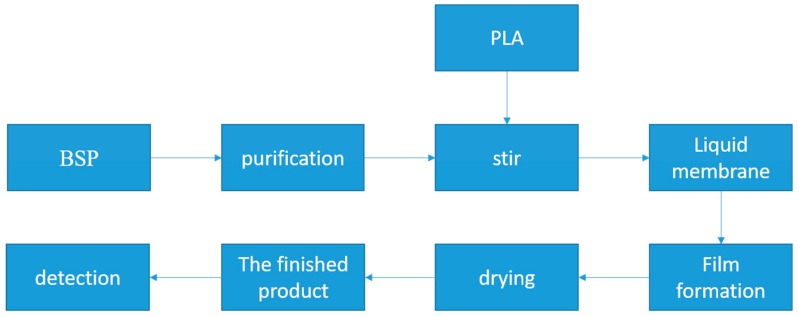
Membrane preparation.

**Table 1 molecules-24-02104-t001:** Glass Transition Temperature of BSP/PLA Composite Films with Different Proportions.

	PLA1	PLA2	PLA3	PLA4	PLA5	PLA6	PLA7	PLA8
Ratio of BSP	0	0.2%	0.4%	0.6%	0.8%	1%	1.2%	1.4%
Tg (℃)	52.1	53.4	55.9	60.9	68.0	55.5	49.9	48.5

**Table 2 molecules-24-02104-t002:** Test Data of BSP/PLA Composite Films.

	Ratio of BSP	T_c_ (℃)	ΔH_c_ (J/g)	T_m_ (℃)	ΔH_f_ (J/g)	X_c_ (%)
PLA1	0	85.5	0.755	167.5	36.32	39.05
PLA2	0.2%	122.3	0.763	167.7	36.89	39.67
PLA3	0.4%	104.2	0.714	167.9	37.71	40.55
PLA4	0.6%	106.3	0.724	168.5	38.02	40.88
PLA5	0.8%	132.2	0.766	169.1	40.68	43.74
PLA6	1%	106.1	0.732	168.3	40.39	43.43
PLA7	1.2%	122.1	0.756	167.9	39.58	42.56
PLA8	1.4%	92.3	0.713	167.7	37.68	40.52

**Table 3 molecules-24-02104-t003:** TG and DTG data of composite films.

	Ratio of BSP	T_initial_ (°C)	T_10_ (°C)	T_complete_ (°C)	T_max_ (°C)
PLA1	0	291.2	232.2	359.3	341.6
PLA2	0.2%	314.5	243.2	371.2	346.9
PLA3	0.4%	318.6	254.3	383.6	349.3
PLA4	0.6%	328.9	286.5	396.7	379.6
PLA5	0.8%	351.2	348.5	398.5	387.5
PLA6	1%	336.8	326.4	396.4	385.8
PLA7	1.2%	332.7	322.1	385.9	352.7
PLA8	1.4%	319.6	285.6	384.2	348.9

**Table 4 molecules-24-02104-t004:** Aperture Size Table of Composite Films.

Number of the Figure	Ratio of BSP (wt%)	Number of Samples	Maximum Aperture (μm)	Minimum Aperture (μm)	Average Aperture (μm)
Figure 6a	0.4	25	2.95	0.22	0.86
Figure 6b	0.6	25	3.21	0.29	1.08
Figure 6c	0.8	25	4.72	0.66	1.51
Figure 6d	1.0	25	1.83	0.81	1.32
Figure 6e	1.2	25	1.88	0.51	1.11
Figure 6f	1.4	25	1.04	0.35	0.93

**Table 5 molecules-24-02104-t005:** Tensile test results of BSP/PLA composite film.

Sample	Ratio of BSP	Width (mm)	Thickness (mm)	Starting Stretch Spacing (mm)	Elastic Modulus (MPA)	Elongation at Break (%)	Tensile Fracture Stress (MPA)	Tensile Strength (MPA)	Tensile Yield Stress (MPA)	Maximum Tensile Force (N)
PLA1	0	10	0.050	25	96.85	21.56	10.12	28.75	0.01	7.26
PLA2	0.2%	10	0.049	25	57.11	26.33	16.90	20.59	9.68	12.56
PLA3	0.4%	10	0.048	25	64.79	25.65	15.21	18.54	8.10	11.89
PLA4	0.6%	10	0.048	25	52.53	35.07	12.54	14.82	3.73	9.56
PLA5	0.8%	10	0.051	25	60.55	53.14	23.18	14.84	9.70	14.38
PLA6	1%	10	0.053	25	52.02	45.73	14.71	15.66	5.94	9.85
PLA7	1.2%	10	0.052	25	67.93	44.81	12.47	24.31	4.70	9.25
PLA8	1.4%	10	0.050	25	57.91	36.53	17.71	19.85	9.96	9.63

**Table 6 molecules-24-02104-t006:** Contact angle data.

	PLA1	PLA2	PLA3	PLA4	PLA5	PLA6	PLA7	PLA8
Ratio of BSP	0	0.2%	0.4%	0.6%	0.8%	1%	1.2%	1.4%
Contact Angle (°)	110.4	103.5	92.6	92.3	88.7	78.2	69.3	53.4

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
