# Peer review of "Preparation and Characterization of Bletilla striata Polysaccharide/Polylactic Acid Composite"

_molecules, 2019, doi:10.3390/molecules24112104_

Round 1

Reviewer 1 Report

The paper reports modification of Polylactic acid (PLA) with Bletilla striata polysaccharide (BSP). PLA is known as a natural polymer with properties making this material interesting for practical applications. Unfortunately, pure PLA exposes some disadvantages: high cost, low heat-resistance, insufficient elasticity, low mechanical strength and low thermal stability. Therefore, modification of PLA and new composite materials preparation is very important task. In the literature there are plenty of papers devoted PLA confirming that this one of the main fields in the science. Therefore, this manuscript concern very challenging task. However, numerous mistakes make it impossible to accept in the current form and require major revision.

Line 99-100: “We speculate that, with increasing BSP content, the agglomerate diameter of BSP particles increases, which hinders the migration and diffusion of PLA chains.” Is it possible to verify somehow this hypothesis? Did you consider DLS or any other method to evaluate size of the particles?

Line 110: Please, add a line with BSP content. Otherwise it is difficult to follow the manuscript because the Experimental part (in this part you explain what PL1 – PL8 are) is at the end of the manuscript. I would recommend also for all figures to keep the same color coding. And this should be also marked in the manuscript or in figure captions.

Line 131: “The positions of the cold crystallization peaks generally shift toward higher temperatures,…” I can observe severe irregularities. How can you explain them. I cannot agree with the trend you mentioned: 85.5, 122.3, 104.2, 106.3, 132.2, 106.1, 122.1, 92.3...

Line 167: You mentioned (lines 100-102) that "In addition, the addition of BSP introduces aromatic heterocyclic rings into the main chain of PLA, which hinders the free rotation of the single bond on the PLA chain, resulting in an increase in the rigidity of the molecular chains of the film." Is it not this statement contradictory with the current one? You introduced additional component (BSP) but it should press PLA chains to decrease distances between them... Please, comment this piece of your paper.

Line 180: What is the small feature at ca. 250°C related to the light green DTG curve?

Line 189-190: “However, the intensity of the diffraction peak first increases and then decreases, with an overall increasing trend” What does it mean? For PL7 and PL8 the total intensity seems to be much smaller and at the level of PL1 (especially PL8). Please, discuss what peaks can be ascribed to PLA and which to BSP. Please, give also Miller indices if possible for the reflections. The positions of the most intense peak seems to vary slightly and especially between PL1 and Pl2 this shift toward smaller 2theta angles is the most significant. Please discuss this effect.

Line 198: It is a pity that you do not supply data for 1, 1.2 or 1.4%. Please, add such data. It seems to be important because in many cases you observe increase with a maximum for 0.8% followed by a decrease. Therefore, I insist you present data for higher (1-1.4%) BSP concentration.

Line 211: “The elastic modulus and tensile strength of the films decrease with the…” Elastic modulus is significantly bigger for PL1 whereas for PL2-PL8 it fluctuates in the broad range. Tensile strength decreases reaching a minimum for PL4 and PL5 and then it rather increases. Please comment your statements.

Line 212: Is it valid also for 1, 1.2 and 1.4? For these composites the crystallinity decreases. Please rewrite this part to be more precise.

Conclusion: please revise this part according to changes and explanations made to other parts of the manuscript

I recommend to revise extensively English.

Language:

Line 29: “excellent application prospects” – please, give more details.

Line 35: “Under the theme of environmental friendly” This sentence is not clear. Please rewrite this sentence.

Line 50-53: instead “Wu et al. prepared a polylactic acid grafted starch/PLA composite. Compared with pure PLA, the obtained composite materials have increased tensile strength…” consider “Wu et al for a polylactic acid grafted starch/PLA composite observed increased tensile strength...”.

Line 63-64: “Feng et al. blended polyurethane elastomers with excellent properties with PLA.” Please, rewrite this sentence.

Line 228: was „11.0 × 10 4” should be „11.0 × 104

Line 235: “After the solution cooled…” should be “After the solution cooling…”

Line 259: “the…” should be “The…”

Author Response

Manuscript ID: molecules-491979

Journal of Molecules

Dear editor,

I have made revisions according to reviewers’ comments. It had been appended below. I hope you can think about my paper again.

Yours sincerely,

Minglong Yuan

Point 1: “We speculate that, with increasing BSP content, the agglomerate diameter of BSP particles increases, which hinders the migration and diffusion of PLA chains.” Is it possible to verify somehow this hypothesis? Did you consider DLS or any other method to evaluate size of the particles?

Response 1: Thanks for your kind suggestions. Based on your suggestions, we discussed it, it was not the particles but the pores. We analysis the pore size later using SEM and NanoMeasurer software. And we reinterpreted the phenomenon. Please, see line 114-118, 212

Point 2: Please, add a line with BSP content. Otherwise it is difficult to follow the manuscript because the Experimental part (in this part you explain what PL1 – PL8 are) is at the end of the manuscript. I would recommend also for all figures to keep the same color coding. And this should be also marked in the manuscript or in figure captions.

Response 2: PLA1, PLA2, PLA3, PLA4, PLA5, PLA6, PLA7, PLA8 and BSP represent the BSP content of 0%, 0.2%, 0.4%, 0.6%, 0.8%, 1%, 1.2%, 1.4% and 100% of the total mass, respectively (line 107-108). And we have corrected the figure again.

Point 3: The positions of the cold crystallization peaks generally shift toward higher temperatures,…” I can observe severe irregularities. How can you explain them. I cannot agree with the trend you mentioned: 85.5, 122.3, 104.2, 106.3, 132.2, 106.1, 122.1, 92.3...

Response 3: After our testing, I am sorry that PLA2 and PLA7 have errors in measurement.

Point 4: You mentioned (lines 100-102) that "In addition, the addition of BSP introduces aromatic heterocyclic rings into the main chain of PLA, which hinders the free rotation of the single bond on the PLA chain, resulting in an increase in the rigidity of the molecular chains of the film." Is it not this statement contradictory with the current one? You introduced additional component (BSP) but it should press PLA chains to decrease distances between them... Please, comment this piece of your paper.

Response 4: I corrected the previous statement: This is because there is a hydrogen bond between BSP and PLA, which hinders the migration and diffusion of the PLA chain, so the molecular chain rigidity increases and the glass transition temperature increases. With the further increase of the BSP ratio, excessive BSP increases molecular chain distance. Therefore, the Tg decreases as the BSP ratio is increased further (line 114-118). When the ratio of BSP is increased further, the distance between the molecular chains is increased, and the entanglement of the PLA chain is reduced, so the thermal stability is reduced. This is consistent with the change in crystallinity (line 175-178).

Point 5: What is the small feature at ca. 250°C related to the light green DTG curve

Response 5: This may be the BSP decomposition first.

Point 6: However, the intensity of the diffraction peak first increases and then decreases, with an overall increasing trend” What does it mean? For PL7 and PL8 the total intensity seems to be much smaller and at the level of PL1 (especially PL8). Please, discuss what peaks can be ascribed to PLA and which to BSP. Please, give also Miller indices if possible for the reflections. The positions of the most intense peak seems to vary slightly and especially between PL1 and Pl2 this shift toward smaller 2theta angles is the most significant. Please discuss this effect

Response 6: The single crystal diffraction peak of 16.69° corresponds to the α crystal form of polylactic acid, and the single crystal diffraction peak of 18.52° is the diffraction peak of BSP. With increasing BSP ratio, the position of the diffraction peak of the PLA single crystal is significantly shifted to a smaller 2θ angle, and then slightly shift to a larger 2θ angle. According to the Brugg equation , it can be found that the crystal plane spacing first increases significantly and then decreases slightly, and the crystal core stacking first loosens and then compacts. Please see line 192-198

Point 7: It is a pity that you do not supply data for 1, 1.2 or 1.4%. Please, add such data. It seems to be important because in many cases you observe increase with a maximum for 0.8% followed by a decrease. Therefore, I insist you present data for higher (1-1.4%) BSP concentration.

Response 7: Through SEM testing, we found that when the concentration is 0.8%, the pores are the largest. Please see line 212

Point 8 and 9: The elastic modulus and tensile strength of the films decrease with the…” Elastic modulus is significantly bigger for PL1 whereas for PL2-PL8 it fluctuates in the broad range. Tensile strength decreases reaching a minimum for PL4 and PL5 and then it rather increases. Please comment your statements.

    Is it valid also for 1, 1.2 and 1.4? For these composites the crystallinity decreases. Please rewrite this part to be more precise.

Response 8 and 9: The elongation at break of the composite films showed a trend of increasing first and then decreasing, reaching the peak value of 53.14% when the BSP ratio is 0.8%. This peak value is 2.46 times greater than that of pure PLA, indicating improved toughness of the material, this is because when the BSP is small, the compatibility of the material is good, and the BSP is uniformly dispersed in the PLA, so the elongation at break increases, when the BSP is large, the partial molecular chain of the PLA may be crosslinked to form a gel point, so the elongation at break will decrease. But the tensile strength of the composite films showed a trend of decreasing first and then increasing, this is because when the BSP is small, the intermolecular interaction force is weak, so the tensile strength is reduced, when the BSP content is high, BSP will crosslink with PLA, and the intermolecular interaction force is strong, so the tensile strength is improved. Please see line214-224

Language:

Point 1: “excellent application prospects” – please, give more details.

Response 1: BSP has high safety as a food additive or ingredient, and its unique properties make it widely used in the food industry. Related research shows that using the excellent film-forming properties of BSP, the preparation of fruit coating film preservative can reduce the evaporation of water and achieve the purpose of preservation. BSP has good anti-inflammatory and acid-resistance ability, and is less affected by factors such as pH, inorganic ions and temperature, and can improve the stability of the emulsified product. With the development and clinical needs of modern pharmaceutical technology, BSP is also widely used in the pharmacology and clinical practice of mediation therapy, coupling agents. Cai et al. developed a white ultrasonic medical couplant, and compared with paraffin oil emulsion and Japanese ultrasonic coupling agent, found that the main quality index of white enamel ultrasonic coupling agent exceeds paraffin oil emulsion, which is superior to Japanese products.

Please see line 29-39

Point 2: Under the theme of environmental friendly” This sentence is not clear. Please rewrite this sentence.

Response 2: The theme of environmental protection and sustainable development

Point 3:  instead “Wu et al. prepared a polylactic acid grafted starch/PLA composite. Compared with pure PLA, the obtained composite materials have increased tensile strength…” consider “Wu et al for a polylactic acid grafted starch/PLA composite observed increased tensile strength...”.

Response: Wu et al. for a polylactic acid grafted starch/PLA composite observed increased tensile strength and elongation at break but have a significantly decreased initial thermal decomposition temperature and deteriorated thermal stability.

Point 4: “Feng et al. blended polyurethane elastomers with excellent properties with PLA.” Please, rewrite this sentence.

Response 4: Feng et al. blended polyurethane with PLA. The obtained materials have the advantages of high elongation at break, high impact strength and high tensile strength but exhibit no improvement in thermal properties.

Point 5: was „11.0 × 10 4” should be „11.0 × 104

Response 5: I have modified this. Please see line 249

Point 6: “After the solution cooled…” should be “After the solution cooling…”

Response 6: I have modified this. Please see line 257

Point 7: “the…” should be “The…”

Response 7: I have modified this. Please see line 281

Reviewer 2 Report

Preparation and characterization of Bletilla striata 2 polysaccharide / polylactic acid composite

Manuscript ID: molecules-491979

The manuscript is presents average research work and can be interesting for readers after following major revision.

1.  Authors need to incorporate some recent reference and reviews related to the presented work to make it more interesting.

2. Page:1 (line 33) author need to add reference related to wide range of application of PLA.

3. Author need to measure contact angle for all the composite samples to find change in hydrophilicity/ hydrophobicity of the samples.

4.  Author need to provide FT-IR for the composite sample with proper labeling of peaks.

5. Methods and materials part should come before results and discussion.

6. Author need to add atleast one application for this composite materials to explain the importance of presented research without that is very difficult to explain the importance of this work.

7. Author need to add graphic/pictorial or schematic presentation  of the work to make it easily understandable.  

8. Quality of Figure 3 need to improve.

9. Page 5 (line: 159) “loss of small amount of impurities” what are those impurities author need to explain.

10. Figure; 4, author need to explain what is the residual weight in TGA profiles.

11. Why only 0.8% (BSP) ration form single crystal.

12. Author need to provide plot for tensile data.

13. SEM-EDS and Elemental analysis for different composite samples is required to establish the composition for all samples.

Author Response

Manuscript ID: molecules-491979

Journal of Molecules

Dear editor,

I have made revisions according to reviewers’ comments. It had been appended below. I hope you can think about my paper again.

Yours sincerely,

Minglong Yuan

Point 1 and 2: Authors need to incorporate some recent reference and reviews related to the presented work to make it more interesting.

Response 1 and 2: Ma et al. obtained the PLA/Fe3O4-AZM microspheres by emulsification-solvent evaporation technology, and the sustained release effect was obvious. Tan et al. obtained the PLA/PCL-PVA-CS-Ag nanofiber auxiliary material by electrospinning technology and achieved good antibacterial effect. Swaroop C et al. used polylactic acid as raw material to prepare magnesium oxide nanoparticle-enhanced biofilm by solvent casting method.

Please see line 47-52

Point 3: Author need to measure contact angle for all the composite samples to find change in hydrophilicity/ hydrophobicity of the samples.

Response 3: please see line 231-234. And I will provide extra figure of it.

Point 4: Author need to provide FT-IR for the composite sample with proper labeling of peaks.

Response 4:

 BSP

PLA1            PLA5

3600-3200 cm-1 has O-H stretching vibration characteristic absorption peak, 3000-2800 cm-1 has C-H characteristic absorption peak, which may be methyl or methine, 1200-1000 cm-1 has C-O-H and C-O-C stretching vibration absorption peak, and 1100-1010 cm-1 range There are 3 absorption peaks indicating that there are glucopyranosyl groups, and 840 wave numbers have absorption peaks indicating that there are α-type glycosidic bonds.

The vicinity of 1456 and 1381 cm-1 is the bending vibration absorption peak of the C-H bond of the methyl group and the methine group of PLA, 1756 cm-1 is the characteristic absorption peak of the carbonyl group on the PLA segment, and 1080 cm-1 is the characteristic absorption peak of the C-O-C characteristic.

The position of the absorption peak of the modified PLA and pure PLA does not change, so there is no change in the functional group, so this is physical modification without any chemical change. The absorption peak around 3400 cm-1 becomes wider and stronger, possibly forming hydrogen bonds

Point 5: Methods and materials part should come before results and discussion.

Response 5: Since this is a journal format requirement, I have to write this way.

Point 6: Author need to add atleast one application for this composite materials to explain the importance of presented research without that is very difficult to explain the importance of this work.

Response 6: In this study, the BSP/PLA composite film was prepared and characterized for the first time. The PLA modified by BSP can be applied to the field of green packaging. Please see line319-321

Point 7:  Author need to add graphic/pictorial or schematic presentation  of the work to make it easily understandable.  

Response 7:

Point 8: Quality of Figure 3 need to improve.

Response 8:

Point 9: Page 5 (line: 159) “loss of small amount of impurities” what are those impurities author need to explain.

Response 9: Through EDS analysis, we found that it was not an impurity, but an unburned BSP and a carbonized PLA. Please see line 167-168

Point 10:  Figure; 4, author need to explain what is the residual weight in TGA profiles.

Response 10:

PLA1

PLA2

PLA3

PLA4

PLA5

PLA6

PLA7

PLA8

BSP

residual weight/%

5.63

0.46

0.84

4.95

3.94

7.15

5.59

4.31

18.73

Point 11: Why only 0.8% (BSP) ration form single crystal.

Response 11: I am sorry that my description of single crystal is wrong. With increasing BSP ratio, the position of the diffraction peak of the PLA single crystal is significantly shifted to a smaller 2θ angle, and then slightly shift to a larger 2θ angle. According to the Brugg equation , it can be found that the crystal plane spacing first increases significantly and then decreases slightly, and the crystal core stacking first loosens and then compacts. Please see line 194-198

Point 12: Author need to provide plot for tensile data.

Response 12:

Point 13: SEM-EDS and Elemental analysis for different composite samples is required to establish the composition for all samples.

Response 13:

The table below represents PLA1, PLA2, PLA3, PLA4, PLA5, PLA6 and BSP from top to bottom. We found that the composite film contains only C, H, O elements and no impurities. And with the increase of BSP, the oxygen content of the film increases.

Element

Weight %

Atomic %

Error %

C K

64.75

76.98

5.49

O K

24.96

22.28

11.54

AuM

10.29

0.75

11.55

Element

Weight %

Atomic %

Error %

C K

59.30

73.32

6.03

O K

27.68

25.69

11.36

AuM

13.03

0.98

11.73

Element

Weight %

Atomic %

Error %

C K

58.75

72.81

6.02

O K

28.17

26.21

11.21

AuM

13.09

0.99

10.18

Element

Weight %

Atomic %

Error %

C K

59.31

72.76

5.94

O K

28.60

26.34

11.22

AuM

12.09

0.90

11.51

Element

Weight %

Atomic %

Error %

C K

60.72

73.00

5.77

O K

29.09

26.25

11.28

AuM

10.19

0.75

11.79

Element

Weight %

Atomic %

Error %

C K

58.94

72.22

5.94

O K

29.24

26.90

11.16

AuM

11.82

0.88

11.73

Element

Weight %

Atomic %

Error %

C K

54.54

66.43

5.98

O K

35.94

32.87

10.55

AuM

9.52

0.71

12.31

Round 2

Reviewer 2 Report

The manuscript is in good shape now and this paper is strongly recommended for publication.